# Research on the Rapid Curing Mechanism and Technology of Chinese Lacquer

**DOI:** 10.3390/polym17121596

**Published:** 2025-06-07

**Authors:** Jiangyan Hou, Tianyi Wang, Yao Wang, Xinhao Feng, Xinyou Liu

**Affiliations:** 1College of Furnishing and Industrial Design, Nanjing Forestry University, Str. Longpan No. 159, Nanjing 210037, China; houjiang@njfu.edu.cn (J.H.); 13861210436@njfu.edu.cn (T.W.); 2381132433@njfu.edu.cn (Y.W.); fengxinhao@hotmail.com (X.F.); 2Co-Innovation Center of Efficient Processing and Utilization of Forest Resources, Nanjing Forestry University, Nanjing 210037, China

**Keywords:** Chinese lacquer, laccase, urushiol, rapid curing, metal ion catalysis, nanomodification, sustainable polymer

## Abstract

Chinese lacquer, a historically significant bio-based coating, has garnered increasing attention in sustainable materials research due to its outstanding corrosion resistance, thermal stability, and environmental friendliness. Its curing process relies on the laccase-catalyzed oxidation and polymerization of urushiol to form a dense lacquer film. However, the stringent temperature and humidity requirements (20–30 °C, 70–80% humidity) and a curing period that can extend over several weeks severely constrain its industrial application. Recent studies have significantly enhanced the curing efficiency through strategies such as pre-polymerization control, metal ion catalysis (e.g., Cu^2+^ reducing drying time to just one day), and nanomaterial modification (e.g., nano-Al_2_O_3_ increasing film hardness to 6H). Nevertheless, challenges remain, including the sensitivity of laccase activity to environmental fluctuations, the trade-off between accelerated curing and film performance, and issues related to toxic pigments and VOC emissions. Future developments should integrate enzyme engineering (e.g., directed evolution to broaden laccase tolerance), intelligent catalytic systems (e.g., photo-enzyme synergy), and green technologies (e.g., UV curing), complemented by multiscale modeling and circular design strategies, to drive the innovative applications of Chinese lacquer in high-end fields such as aerospace sealing and cultural heritage preservation.

## 1. Introduction

The Chinese lacquer tree (*Toxicodendron vernicifluum*), a member of the Anacardiaceae family, has been cultivated for millennia across East Asia for its sap-derived coating material [1]. To harvest raw lacquer, mature trees (typically 10–15 years old) are carefully tapped during the summer months by making diagonal incisions in the bark. The exuded milky sap, composed of urushiol (60–65%), water (20–30%), plant gums, glycoproteins, and enzymes, undergoes natural oxidation to form the characteristic durable coating [2]. This labor-intensive harvesting process, requiring skilled artisans to collect and refine the sap while avoiding dermatitis caused by urushiol, contributes to the material’s historical value and production limitations [3].

Chinese lacquer stands as a sustainable bio-based coating renowned for exceptional corrosion resistance, thermal stability, and eco-friendly attributes [2]. Its curing mechanism depends on the oxidative polymerization of urushiol (Figure 1), a catechol derivative with predominantly C15 side chains containing 1–3 unsaturated bonds [4]. The most abundant urushiol homolog (≈70% of total content) features three conjugated double bonds in its alkenyl side chain (3-[pentadeca-8′,11′,14′-trienyl]-catechol), which critically determines the polymerization kinetics and final coating properties [5].

Notably, the urushiol composition exhibits significant geographic and climatic dependence. Studies demonstrate that trees growing in warmer regions (e.g., southern China) produce urushiol with higher degrees of side chain unsaturation compared to temperate zone counterparts [5]. Seasonal variations in rainfall and temperature further influence the ratio of monoene/diene/triene side chains, with summer-harvested lacquer containing 15–20% more triene derivatives than autumn batches [6]. These compositional differences directly impact the curing speed and film hardness, necessitating strict origin control for industrial applications [7].

The curing process involves laccase-catalyzed oxidative polymerization requiring precise environmental control (20–30 °C, 70–80% RH) to maintain enzymatic activity [7]. Recent advances have explored accelerated curing through pre-polymerization techniques and nanomaterial additives. However, challenges persist in scaling these methods while preserving the material’s unique aesthetic and mechanical properties.

This review systematically analyzes the chemical mechanisms underlying rapid curing strategies, evaluates emerging bio-catalytic approaches, and proposes sustainable engineering solutions to bridge traditional craftsmanship with modern manufacturing requirements.

## 2. Curing Mechanism and Influencing Factors of Chinese Lacquer

The primary chemical constituents of Chinese lacquer include urushiol (50–70%), water (30–40%), laccase, polysaccharides, glycoproteins, and other organic substances [8]. The film-forming capability originates from urushiol, a mixture of catechol derivatives with C15–C17 unsaturated alkyl side chains (general formula C_6_H_3_(OH)_2_R, R = (CH_2_)_14_CH_3_ or analogs). These ortho-dihydroxyphenyl structures enable dual curing pathways: enzymatic oxidation catalyzed by laccase and auto-oxidative polymerization driven by side chain unsaturation. The crosslinking density critically depends on the length and degree of unsaturation of the alkyl side chain—longer triene chains (e.g., C17:3) significantly enhance radical propagation due to greater delocalization and flexibility, while shorter or saturated chains exhibit lower reactivity and hinder network formation [9,10].

Laccase (EC 1.10.3.2), a thermolabile blue multicopper oxidase with a T1/T2/T3 copper cluster, initiates curing through a four-electron transfer mechanism. At optimal conditions (pH 6.5–7.0, 25–30 °C), it oxidizes urushiol’s phenolic groups to semiquinone radicals (detectable by ESR at λmax = 420 nm), followed by electron transfer to molecular oxygen, which is reduced to water at the T2/T3 active site. Component extraction methods significantly influence system reactivity: traditional cold-pressing (<40 °C) preserves native laccase activity but yields heterogeneous urushiol, while acetone/ethanol fractionation coupled with ion-exchange chromatography produces >95% pure urushiol with retained polymerization capacity. Advanced membrane separation (50 kDa ultrafiltration) concentrates laccase with 85% activity retention by removing inhibitory polysaccharides [11].

Importantly, while raw urushiol is a known sensitizer capable of inducing urushiol-induced contact dermatitis, its allergenic activity is markedly reduced upon polymerization. Studies have shown that the fully cured lacquer film forms a dense crosslinked matrix in which residual free urushiol is below the detectable thresholds or rendered non-reactive, thus substantially mitigating skin sensitization risks under normal conditions of use [12]. The aging kinetics of Chinese lacquer have also been investigated, primarily through spectroscopic and chromatographic methods. Over time, the unreacted urushiol content decreases significantly within the first 7–14 days of curing, correlating with increasing crosslink density. Long-term aging under ambient conditions shows negligible further urushiol release or degradation, indicating high chemical stability of the polymerized film [13]. This mechanistic understanding enables targeted strategies to accelerate curing while preserving the ecological superiority of this millennia-old biomaterial.

### 2.1. Curing Mechanism of Chinese Lacquer

The curing mechanism of raw lacquer involves both enzyme-catalyzed and non-enzymatic reactions, with enzymatic reactions playing the dominant role, while non-enzymatic reactions are key to forming the final high-molecular-weight network structure that largely determines the physical and mechanical properties of the lacquer film [14]. Studies have shown that lacquer liquids with higher degrees of enzymatic polymerization exhibit more readily oxidized side chains, and in low-humidity environments, the monomer content of urushiol in the cured film can be reduced to below 27% [15].

The film-forming process of raw lacquer critically depends on the catalytic action of laccase. As a multi-copper oxidase, laccase oxidizes polyphenolic compounds, generating free radicals that trigger cross-linking and thereby accelerate the curing process. Before application, raw lacquer is typically heated (~40 °C) and stirred to reduce its water content to 5–10%. During this pretreatment, partial pre-polymerization occurs as the lacquer is exposed to air [16,17].

Structurally, laccase (Figure 2) coordinates four copper ions through three distinct binding motifs: the Type I (T1) copper (green region) features a distorted tetrahedral geometry with two histidine imidazoles, a cysteine thiolate, and a methionine thioether, conferring intense blue absorption (ε614 ≈ 5000 M^−1^cm^−1^) and enabling direct substrate oxidation via outer-sphere electron transfer. The trinuclear T2/T3 cluster (yellow/blue regions) consists of a mononuclear Type II (T2) copper in square planar H_2_O/His coordination and a binuclear Type III (T3) site bridged by a hydroxyl group, forming an antiferromagnetically coupled (J ≈ −150 cm^−1^) oxygen-binding pocket [18].

Spatial organization places the T1 site 12–13 Å from the T2/T3 cluster, connected via a conserved His-Cys-His pathway for electron relay. During urushiol oxidation, T1-Cu^2+^ abstracts electrons from phenolic hydroxyls, generating semiquinone radicals while reducing to T1-Cu^+^. Concurrently, the T2/T3 cluster activates O_2_ through concerted action: T2-Cu^2+^ polarizes O_2_ as a Lewis acid, while the T3 μ-hydroxide bridge donates electrons, ultimately reducing O_2_ to two H_2_O molecules. Radical coupling specificity arises from T1′s hydrophobic substrate pocket (Phe/Leu residues), which orients urushiol’s triene chain perpendicularly to favor C8-C8′ dimerization, though ortho-C and C-O couplings contribute to oligomer diversity. Subsequent three-dimensional network formation relies on the T2/T3 cluster’s rapid redox cycling (kcat ≈ 10^2^−10^3^ s^−1^) and residual T1-Cu^+^-mediated allylic C-H activation (BDE ≈ 85 kcal/mol), propagating radicals through unsaturated side chains. Notably, T2-depleted laccases retain the T1/T3 geometry but exhibit 90% activity loss due to increased O_2_ activation energy (λ ≈ 1.2 eV), confirming T2′s role in electronic coupling rather than structural stabilization [19].

The curing process of Chinese lacquer can be divided into three main stages (Figure 3):

Oxidation of Urushiol: Under appropriate temperature and humidity, the Cu^2+^ ions in laccase catalyze the oxidation of the ortho- and para-hydroxyl groups of the phenol rings, generating highly reactive semiquinone radicals, which then undergo dismutation to form lacquer quinones. During this process, Cu^2+^ is reduced to Cu^+^, which is subsequently reoxidized by oxygen to regenerate Cu^2+^, maintaining the catalytic cycle [20].

Oligomer Formation: The lacquer quinones and semiquinone radicals generated in the initial oxidation step serve as reactive intermediates for subsequent coupling reactions. These species engage in multiple types of radical-mediated coupling pathways, including the following:C–C coupling at the meta- or para-positions of aromatic rings to form biphenyl- or diaryl-type dimers, e.g., C8–C8′, C8–C5′ linkages.C–O coupling, where semiquinone radicals undergo oxidative etherification at ortho-positions.Side chain coupling, involving the reaction of unsaturated triene side chains with aromatic radicals, forming allyl-aryl linkages or extended conjugated networks.

These mechanisms result in a structurally diverse set of dimers and oligomers—over 20 representative structures have been identified [21]. For instance, urushiol derivatives can form both linear and branched oligomers depending on the coupling sites and radical lifetimes. The highly unsaturated side chains provide multiple reaction sites, enabling simultaneous aromatic and aliphatic polymer growth. These intermediate oligomers are crucial precursors for the next stage of film formation.

Three-dimensional Network Formation: The oligomers formed via the above mechanisms undergo further polymerization into longer chains. Their unsaturated side chains participate in self-condensation or radical-induced cross-linking reactions, ultimately forming a robust three-dimensional network. This process results in a hardened film composed of complex, high-molecular-weight polymers, characterized by a dense, thermoset-like structure.

Moreover, the radical chain polymerization mediated by laccase also plays a significant role. The generated radicals can add to surrounding urushiol molecules or unsaturated groups, forming extended conjugated structures that propagate throughout the system. Importantly, this process depends not only on laccase activity but also on the structural characteristics of the substrate, indicating that tailoring the catalytic system to specific urushiol structures is crucial for enhancing curing efficiency [22,23].

### 2.2. Influencing Factors in Lacquer Curing

Zhao et al. [24] reported that the recombinant laccase rSC-2 from myxobacteria exhibited optimal activity at 60 °C and pH 7.0. Low concentrations of Cu^2+^ and Mn^2+^ enhanced laccase activity, whereas high concentrations of Fe^3+^, Co^2+^, and Ba^2+^ inhibited its catalytic function. Although this study did not directly focus on lacquer phenol substrates, it indicates that temperature, pH, and the types and concentrations of metal ions significantly influence laccase activity, and thus optimizing reaction conditions is critical for accelerating lacquer curing.

#### 2.2.1. Temperature

Temperature is a key factor influencing laccase activity. As a biological catalyst, laccase exhibits maximal activity within an optimal temperature range, and deviations either way can cause activity loss or even denaturation. Pawlik et al. [25] found that culturing Cerrena unicolor at 4 °C upregulated laccase gene expression, though with limited enhancement of enzymatic activity, while treatment at 40 °C inhibited gene transcription but maintained enzyme activity through post-translational modifications. Additionally, high temperatures (above 30 °C) can disrupt non-covalent interactions such as hydrogen bonds and hydrophobic forces, leading to conformational destabilization and decreased activity [26], whereas lower temperatures, although slowing reaction rates, help preserve the enzyme structure [27]. Thus, controlling the temperature during natural lacquer curing is crucial for maintaining laccase activity and promoting rapid, stable film formation.

#### 2.2.2. pH

The pH also significantly impacts laccase activity. Jean-Marc Bollag et al. [28] demonstrated that laccases from different fungi exhibit distinct optimal pH ranges (Table 1): for instance, Rhizoctonia praticola shows peak activity at neutral pH, whereas *Fomes annosus*, *Pholiota mutabilis*, and *Trametes versicolor* prefer acidic conditions. Galhaup et al. [29] further revealed that laccase activity toward phenolic substrates is optimal at pH 3–4.5, while activity toward non-phenolic substrates favors an even lower pH.

At the molecular level, the pH affects laccase activity by modulating the redox potential of substrates, the protonation state of the T2 copper site, and the rate of electron transfer [30]. Recent work on local environment engineering (e.g., amino acid mutations) has successfully shifted the optimal pH range, providing a theoretical foundation for rational enzyme engineering [31].

The enzymatic activities listed in Table 1 were determined using standard spectrophotometric assays by monitoring the oxidation rate of each substrate at its corresponding optimal pH, with absorbance measured at specific wavelengths characteristic of the oxidation products. Activities are expressed in relative units, normalized to the activity based on 2,6-dimethoxyphenol, which is set as 1000. pH values indicate the pH at which maximum activity was observed for each substrate.

From an application standpoint, fine-tuning the pH of the curing system not only enhances laccase catalytic efficiency but also effectively shortens the curing time of natural lacquer.

#### 2.2.3. Other Factors

Proteins and polysaccharides also play important roles in affecting laccase activity. The glycoproteins in raw lacquer consist of approximately 10% carbohydrates and 90% proteins and contain 15 types of amino acids [32]. Although glycoproteins do not catalyze lacquer phenol oxidation themselves, they exhibit a certain inhibitory effect on laccase activity. Due to their extremely low solubility, the structural and functional characteristics of lacquer glycoproteins remain poorly understood.

In addition, inorganic pigments commonly added to colored lacquer, such as red lead (Pb_3_O_4_) and titanium dioxide (TiO_2_), may influence the drying rate of raw lacquer or induce photochemical reactions, thereby altering the film-formation process and durability of the lacquer film [33].

## 3. Strategies for Accelerating the Curing of Chinese Lacquer

### 3.1. Catalytic Curing Mechanism of Metal Ions

#### 3.1.1. Catalytic Mechanism and Effects of Copper Ions

Copper ions have emerged as highly effective catalysts in accelerating the curing of natural lacquer. In a study by Huang Ting and colleagues [34], a Cu-D370 composite was synthesized by combining copper ions with a weakly basic styrene-based anion exchange resin (D370), thereby emulating the catalytic behavior of natural laccase (a copper-containing polyphenol oxidase). Their results, summarized in Table 2, demonstrated that the lacquer film catalyzed by the Cu-D370 complex achieved a dry-to-touch state within one day, significantly faster than the three days required for films catalyzed by immobilized laccase, referred to as “super urushi”.

However, while both the Cu-D370-catalyzed film and super urushiol polymer showed superior drying rates, gloss, and impact strength compared to ordinary urushi, a notable reduction in adhesion was observed (grade 1–2 vs. grade 4). This decrease in adhesion can be attributed to the rapid curing process, which may hinder sufficient interfacial interactions between the lacquer and substrate. Faster polymerization can result in a denser surface layer forming prematurely, thereby limiting the diffusion and penetration of the urushiol monomers into the substrate. Additionally, the altered film morphology and increased surface smoothness may reduce mechanical interlocking, further weakening the adhesion performance.

Building on the understanding of biomimetic catalysis, Honzíček et al. [35] investigated synthetic analogs, such as cashew nutshell liquid derivatives, and found that, when coordinated with iron–salen complexes, these materials could replicate the laccase-like oxidative polymerization of phenols in the presence of hydrogen peroxide.

Further advancing this field, Zheng Lu [36] identified that silver nitrate (AgNO_3_) could initiate the oxidation of lacquer phenols via a single-electron-transfer mechanism mediated by Ag^+^ ions, leading to the generation of semiquinone radicals that subsequently undergo radical coupling to form polymers (PUL). Similar oxidative polymerization mechanisms involving copper ions (Cu^2+^) have also been reported. In parallel, Lavigne et al. [37] demonstrated that oxygen-bridged copper(II) complexes, such as (pyridine)_4_Cu_4_Cl_4_O_2_, could promote phenoxy radical formation through a single-electron-transfer process, effectively triggering coupling and cross-linking reactions.

Collectively, these studies highlight the crucial role of copper ions in enhancing the oxidative cross-linking of lacquer phenols—the key reactive component of Chinese lacquer. The catalytic process primarily involves copper ions coordinating with hydroxyl groups and double bonds within phenol molecules, facilitating electron transfer and radical generation, thus promoting intermolecular cross-linking. This mechanism not only accelerates the curing process but also significantly improves the mechanical strength and chemical resistance of the resulting lacquer films.

Nevertheless, an excessive concentration of copper ions may lead to side reactions, such as the over-oxidation of phenolic components, adversely impacting the final film quality [38]. Consequently, precise control over the copper ion concentration is essential to optimize the catalytic performance while avoiding detrimental effects.

Despite these challenges, the use of copper ions introduces additional concerns related to the cost and environmental impact. In response, researchers have been exploring more sustainable catalytic systems. For example, Zhang Yan et al. [39] developed a copper-based nanozyme by coordinating copper ions with ethanolamine (Figure 4), mimicking the catalytic oxidation behavior of natural laccase. Their experiments revealed that mixing the Cu-ethanolamine nanozyme with raw lacquer at a 1:20 ratio significantly shortened the drying time while maintaining the industrial performance standards required for lacquer films. Notably, these novel catalysts retain the high catalytic efficiency of copper ions while reducing toxicity and environmental hazards.

#### 3.1.2. Catalytic Effects of Other Metal Ions

Beyond copper ions, other transition metals have also been investigated for their catalytic effects on lacquer curing. Zheng Xiaoxiao and collaborators [40] successfully synthesized a Fe^3+^-based metal-organic framework (MIL-101(Fe)-3) capable of catalyzing the oxidative polymerization of deactivated lacquer (commonly referred to as “dead lacquer”). Remarkably, film formation was achieved within 8 h at ambient temperature, compared to uncatalyzed lacquer, which remained uncured even after 1440 h.

In another study, Zhang Li et al. [41] conducted a systematic investigation into the impact of 11 different metal ions on the absorption spectra during laccase-catalyzed substrate oxidation. Their findings indicated that diamagnetic ions such as Zn^2+^, Cd^2+^, Ca^2+^, and Mg^2+^ enhanced the kinetic stability of the reaction products, thus facilitating the curing process. Conversely, Fe^2+^ exhibited a notable inhibitory effect, attributed to its ability to reduce semiquinone radicals back to their substrate form, temporarily suppressing the curing reaction. This inhibition persisted until the Fe^2+^ ions were fully oxidized to Fe^2+^, after which normal catalytic activity resumed. Further investigations by Yu Zongping et al. [42] revealed that Co^2+^, Mn^2+^, and Zn^2+^ ions could bind to oxygen to form peroxides, subsequently generating radicals that accelerated the curing reaction. Among these, cobalt ions displayed particularly strong catalytic activity, promoting radical generation through a redox cycle between Co^2+^ and Co^3+^.

Despite the promising performance of these alternative metal ions, copper ions continue to outperform in terms of catalytic activity and versatility. Comparative studies have consistently shown that copper-based catalysis exhibits significantly higher efficiency, especially under low-temperature conditions [43], offering new possibilities for the rapid curing of lacquer under complex environmental conditions.

### 3.2. Repeated Kneading Mixing Technology

To accelerate the enzymatic polymerization of raw lacquer sap, researchers developed a small kneading mixer (Figure 5). This device allows for the repeated kneading and stirring of the lacquer sap, a process known as “kurome” in Japanese. Through repeated kneading, the transformation of lacquer monomers into dimers, trimers, tetramers, and other oligomers and polymers is effectively promoted [44].

Specifically, the raw lacquer sap is initially stirred in an open vessel at room temperature for approximately 1.5 h. The temperature is then gradually increased to 40 °C and maintained for 2–4 h until the moisture content decreases to 3–5%, producing the first kneading sample, K-0. Water is subsequently added to K-0 to restore the moisture content to about 20–25%, followed by another round of kneading. By repeating this process, samples K-1, K-2, K-3, and K-4 are obtained, where K-0 undergoes only one refining step, and K-1 to K-4 correspond to one to four rounds of repeated kneading, respectively.

After each kneading cycle, the molecular weight distribution of the samples is analyzed at 40 °C using aqueous-phase gel permeation chromatography (GPC). The results show that with an increasing number of kneading cycles, the monomer content in the lacquer sap decreases, while the content of oligomers and polymers increases, confirming the ongoing enzymatic polymerization during the repeated kneading process.

Additionally, the antioxidative properties of the raw and kneaded lacquers were evaluated using thermogravimetric analysis (TG) and derivative thermogravimetry (DTG). The results indicate that as the number of kneading cycles increases, the antioxidative capacity of the lacquer sap gradually decreases, the autoxidation of side chains is enhanced, and the drying time of the lacquer film is significantly shortened.

### 3.3. Hybrid Lacquer Technology

The liquid component of lacquer sap, urushiol, forms a durable lacquer film through laccase-catalyzed polymerization. During this enzymatic process, both the activity of laccase and the antioxidative capacity of the phenolic structure play critical roles. Higher laccase activity accelerates the polymerization reaction, significantly reducing the drying time of the lacquer film. Conversely, the lower antioxidative capacity of phenolic components can promote the autoxidation of olefins in the side chains. Although the direct enhancement of laccase activity is challenging, it is possible to reduce the antioxidative capacity of urushiol by introducing chemical reagents such as silanes or organosilicon compounds [45]. These reagents react with the hydroxyl groups of urushiol molecules, thereby lowering the concentration of urushiol monomers. Based on this approach, researchers have long focused on developing hybrid lacquers using raw lacquer sap and kurome-processed lacquer.

Considering that organosilicon compounds containing chlorine or thiol (–SH) groups may inhibit laccase activity, various silanes free of chlorine or thiol residues were selected for hybrid lacquer preparation. Furthermore, the reaction mechanisms between urushiol and organosilane were investigated [46].

Analytical techniques, including infrared spectroscopy (IR), nuclear magnetic resonance (NMR), and thermogravimetric analysis/differential scanning calorimetry (TGA/DSC), were employed to study the reaction products. The results revealed that silanes containing amino groups (e.g., N-(β-aminoethyl)-γ-aminopropyltrimethoxysilane) or epoxy groups (e.g., β-(3,4-epoxycyclohexylethyl)trimethoxysilane) reacted rapidly with urushiol through an alcoholysis reaction, significantly promoting polymerization. Additionally, amino residues were found to induce ring-opening reactions of epoxy groups, forming an interpenetrating network structure.

Since the alcoholysis reaction occurs between the alkoxy groups of organopolysiloxane and the hydroxyl groups of urushiol, it is speculated that a three-dimensional cross-linked polymer film is formed through repeated Si–O bond formation. The reaction processes involving urushiol and organosilanes, including both the alcoholysis and amino-induced mechanisms, are illustrated together in a consolidated schematic (see Figure 6).

Based on the hybrid lacquer research results, it was concluded that the addition of organosilanes primarily facilitates reactions at the hydroxyl groups of urushiol molecules. In the lacquer sap, free water first reacts through sol-gel processes to form silyloxy residues. Subsequently, the hydroxyl groups of urushiol are attacked to generate urushiol radicals, promoting continuous polymerization and eventually forming a lacquer film [44].

Test results further indicated that other components in the lacquer sap, such as polysaccharides and glycoproteins, do not react with the organosilane. Typically, lacquer film formation relies on the oxidation of urushiol catalyzed by laccase to generate free radicals, followed by coupling reactions and autoxidation along the side chains. However, because laccase activity requires specific temperature and 70–80% relative humidity conditions, the drying process for traditional natural lacquer demands special drying chambers, making process management difficult.

By mixing lacquer sap with organosilane, a chemical reaction occurs between urushiol and silane compounds, endowing the resulting hybrid lacquer with fast-drying properties. It is therefore anticipated that such hybrid lacquers will find applications not only in traditional crafts but also in outdoor industrial fields. Recently, research teams reported the application of hybrid lacquers as chromium-free corrosion-resistant coatings, demonstrating that urushiol-silicate compounds provide corrosion protection comparable to conventional chromate treatments [47].

### 3.4. Nanoparticle Modification Technology

It is well known that smaller particle sizes correspond to larger specific surface areas. In a lacquer emulsion, reducing the droplet size increases the contact area between urushiol and laccase, thereby facilitating enzymatic reactions and accelerating film drying. Based on this principle, researchers utilized a laboratory-built small kneading mixer to process raw lacquer (Figure 4) and measured the droplet size using a particle size analyzer.

Optimal dispersion was achieved at 240 rpm stirring for 60 min, yielding an average droplet size of 0.194 μm. Under other conditions, average sizes were recorded as 0.233–0.337 μm. The finer droplet dispersion significantly enhanced the lacquer’s color intensity, gloss, and UV durability. Traditionally, colored lacquers are prepared by adding organic or inorganic pigments; however, some pigments, such as cinnabar (HgS), realgar (As_4_S_4_), and red lead (Pb_3_O_4_), are toxic or interfere with film drying. Others, like titanium dioxide (TiO_2_), may trigger photochemical degradation.

Moreover, conventional organic dyes, due to their conjugated structures, are prone to photodegradation, resulting in poor lightfastness. Recently, a novel coloring method using noble metal nanoparticles has been developed. Researchers prepared nanoscale lacquers incorporating gold and silver nanocolloids (particle sizes ~15 nm and 5–8 nm, respectively) provided by Nippon Paint Co, Hongkong, China [48].

These colloids dispersed uniformly within the lacquer, producing vivid colors: gold colloids yielded a red tone (similar to standard code S41-142), while silver colloids imparted a yellow tone (similar to code S44-257) [49]. The resulting nano-lacquers demonstrated superior lightfastness after microdispersion processing and colloid incorporation.

This fast-drying, nanoparticle-enhanced lacquer—integrating organosilanes, polyurethanes, epoxy resins, and noble metal colloids—is expected to find broader applications beyond traditional crafts, particularly in industrial outdoor coatings due to its excellent UV and water resistance.

The incorporation of metal nanoparticles into coatings is well-established [50], with their significantly larger specific surface areas compared to bulk metals [51]. Morphologically, metal nanoparticles exhibit diverse forms—including sheets, tubes, cubes, rods, triangles, and polyhedra [52]—which expose more reactive surface sites. For instance, 10 nm cobalt nanoparticles possess a surface area thousands of times greater than that of cobalt ions of an equivalent mass, dramatically enhancing their interaction with active lacquer components like urushiol.

Hanxing Wang et al. [53] further demonstrated that adding nano-Al_2_O_3_ modified with 3-aminopropyltriethoxysilane (KH550) effectively shortened the lacquer drying time. With 4% nano-Al_2_O_3_ addition, the film’s hardening time at 25 °C and 80% humidity decreased from 210 min to 120 min. Nano-Al_2_O_3_ accelerated the oxidative polymerization of urushiol, promoted cross-linked network formation, and filled film pores, resulting in enhanced compactness. Consequently, film hardness increased from 3H to 6H, impact resistance improved from 40 kg × cm to 50 kg × cm, and hydrophobicity was enhanced due to increased surface roughness.

## 4. Key Challenges in Chinese Lacquer Curing

The curing of Chinese lacquer faces several critical challenges across multiple dimensions. Laccase, the core catalyst driving urushiol polymerization, operates within a narrow range of optimal conditions (20–40 °C, pH 6.0–7.5), making it highly sensitive to environmental fluctuations and limiting its adaptability to industrial settings. The enzymatic activity rapidly deteriorates outside these conditions; for instance, high-temperature curing at 80 °C shortens the laccase half-life to less than 72 h [54]. Additionally, interference from metal ions such as Fe^2+^, which reduces free radicals, further complicates catalytic system design. Balancing the curing efficiency and film quality presents another significant hurdle: while accelerated curing methods, including Cu^2+^ catalysis and nanoparticle incorporation, can speed up film formation, they also risk inducing over-oxidation and structural defects that compromise durability. Pre-polymerization techniques have shown promise in reducing curing times by 30–50%, yet they introduce greater monomer heterogeneity, leading to inconsistent mechanical properties across the lacquer film. Scalability and cost considerations remain substantial obstacles as well, with high energy demands—such as the need for 40 °C thermal catalysis during kneading or microfluidic reactor operation—hindering cost-effective production. Furthermore, nanoparticle dispersion instability (evidenced by zeta potentials below ±20 mV) and aggregation issues during upscaling exacerbate material waste. Sustainability concerns also persist: the use of toxic pigments such as HgS and Pb_3_O_4_ undermines the eco-friendly claims of bio-based lacquer, while VOC emissions generated during hybrid lacquer synthesis (e.g., organosilane reactions) pose challenges for achieving green certification. Finally, knowledge gaps remain regarding non-enzymatic pathways, particularly in understanding the interactions between glycoproteins and polysaccharides in stabilizing laccase activity, as well as the ambiguous mechanisms governing the autoxidation of unsaturated side chains under low-humidity conditions.

## 5. Future Research Priorities

Future research on Chinese lacquer curing should prioritize several key directions to overcome current limitations and unlock broader applications. Advanced enzyme engineering is essential, focusing on developing laccase mutants through directed evolution or rational design to expand pH and temperature tolerance—such as T2 copper site modifications that enhance oxygen reduction efficiency—and constructing immobilized enzyme systems (e.g., metal-organic framework carriers) to achieve operational stability exceeding 200 h. Smart catalytic systems offer another promising avenue, including the design of stimuli-responsive catalysts (such as photoactivated Cu-nanozymes and pH-switchable Fe^3+^ complexes) for the spatiotemporal control of polymerization, as well as hybrid systems that combine laccase with abiotic catalysts (e.g., laccase-Ag^+^ co-catalysis) to decouple radical generation from environmental constraints. Green process intensification should also be pursued, optimizing low-energy curing methods like UV/visible light-assisted polymerization at temperatures below 50 °C or atmospheric plasma activation for surface treatment, alongside the development of bio-based alternatives to toxic pigments, such as plant-derived anthocyanin–nanoTiO_2_ composites. Furthermore, multi-scale computational modeling will be critical for accelerating innovation, with machine learning models (targeting R^2^ > 0.95) predicting the curing kinetics based on variables like urushiol side chain unsaturation, the nanoparticle size, and humidity-time coupling, complemented by molecular dynamics simulations to reveal glycoprotein–laccase interfacial mechanisms. Integrating circular economy principles, research should focus on valorizing lacquer waste via enzymatic depolymerization to recover urushiol at yields above 80% and designing modular lacquer composites for easy disassembly and reuse, particularly in aerospace and heritage restoration. Finally, for successful industrial translation, it will be crucial to establish ASTM/ISO-compliant protocols for evaluating accelerated aging resistance (e.g., QUV and Corrodkote tests) and curing energy efficiency and to collaborate with coating manufacturers to pilot hybrid lacquer formulations in demanding sectors, such as automotive primers and marine anticorrosion coatings.

## 6. Conclusions

Chinese lacquer, an ancient yet sustainable bio-coating, demonstrates unparalleled potential as an eco-friendly material due to its corrosion resistance, thermal stability, and cultural significance. Its curing process, governed by the laccase-catalyzed oxidative polymerization of urushiol, requires precise environmental control (20–30 °C, 70–80% humidity), posing challenges for industrial scalability. Recent advances in pre-polymerization, nanomaterial integration, and hybrid technologies have markedly accelerated the curing efficiency—Cu^2+^ catalysis reduces drying time to one day, while nanoparticle modifications enhance UV resistance and mechanical strength. However, persistent hurdles include laccase’s environmental sensitivity (e.g., activity loss at >40 °C), trade-offs between curing speed and film durability, and sustainability concerns from toxic additives (HgS, Pb_3_O_4_) and VOC emissions. Future innovation must prioritize enzyme engineering to broaden laccase’s operational range, develop stimuli-responsive catalysts (e.g., photoactivated nanozymes), and integrate circular economy principles for waste valorization. By synergizing biocatalysis, computational modeling, and green process optimization, Chinese lacquer can transcend its traditional roots to address modern industrial demands, offering sustainable solutions for aerospace coatings, heritage conservation, and beyond. Bridging lab-scale breakthroughs with scalable, eco-certified production will ultimately unlock its full potential as a 21st-century advanced material.

## Figures and Tables

**Figure 1 polymers-17-01596-f001:**
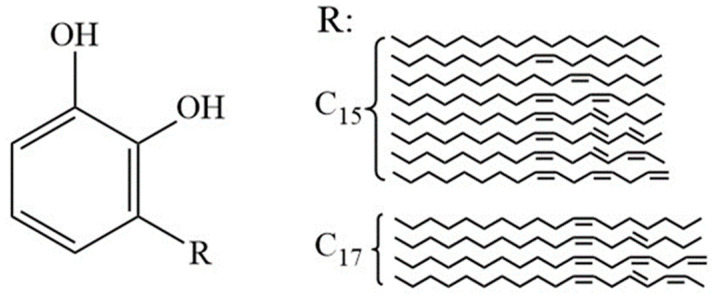
Structure and content of urushiol in Chinese lacquer.

**Figure 2 polymers-17-01596-f002:**
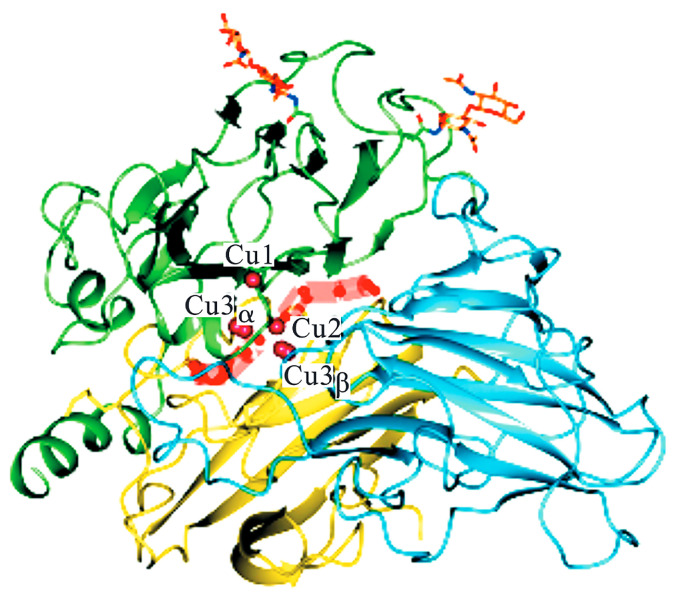
Copper ion binding sites of laccase [18].

**Figure 3 polymers-17-01596-f003:**
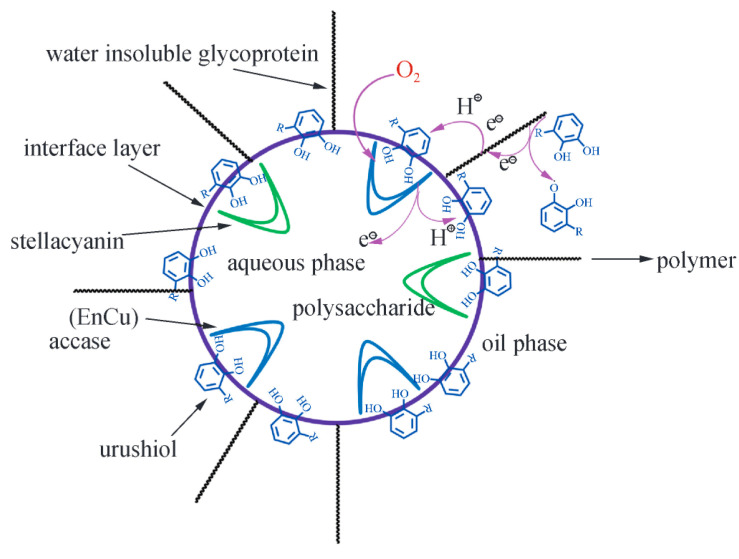
Schematic of polymerization and curing of Chinese lacquer phenol interface.

**Figure 4 polymers-17-01596-f004:**
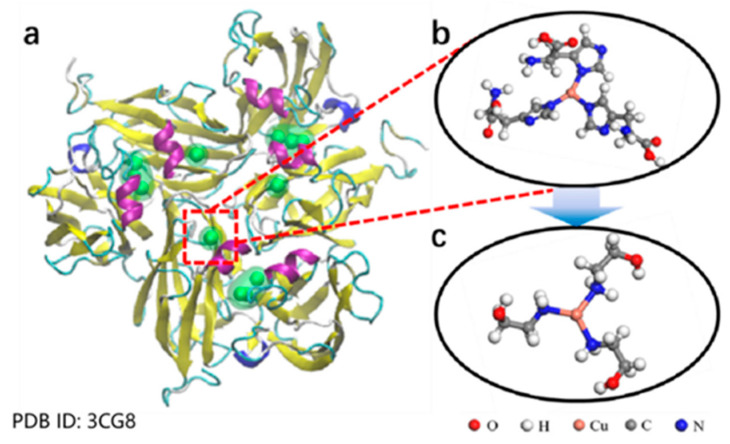
(**a**) Structure of laccase. (**b**) Enlarged view of the structure of copper active centers. (**c**) Schematic diagram of Cu-ethanolamine nanozyme structure [39].

**Figure 5 polymers-17-01596-f005:**
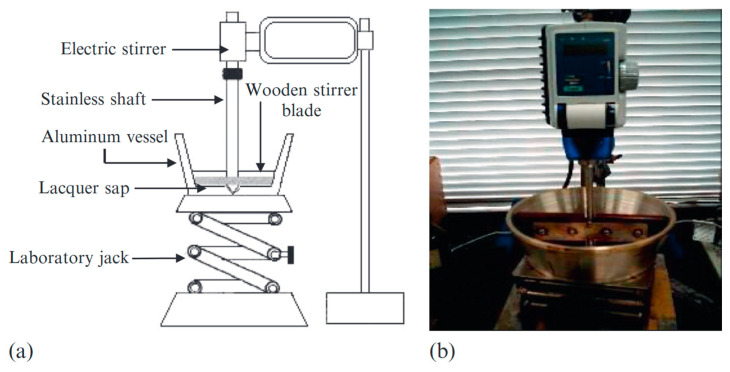
Image (**a**) and photo (**b**) of kneading mixer [8].

**Figure 6 polymers-17-01596-f006:**
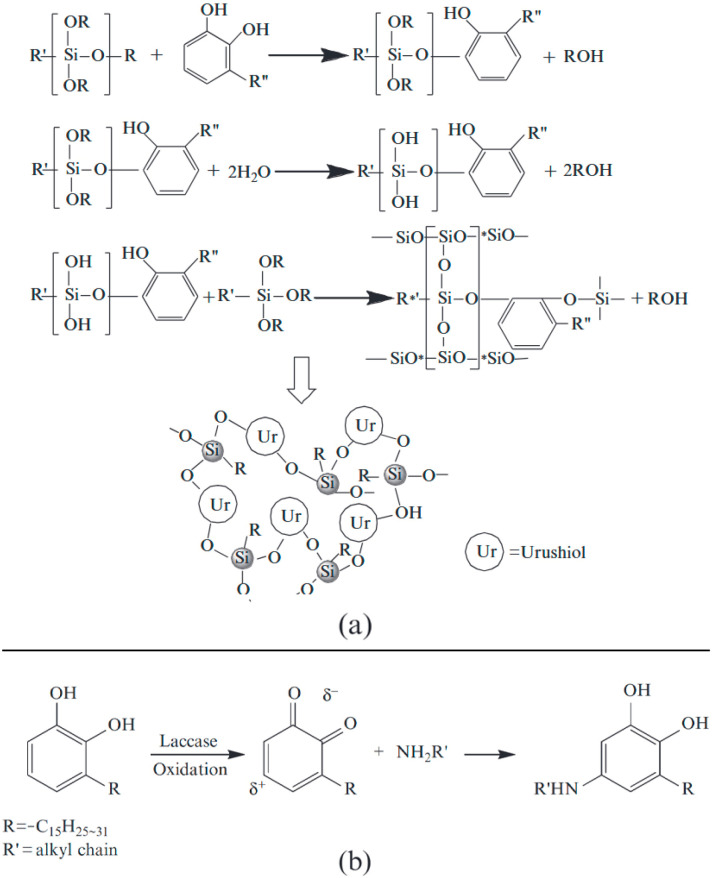
Schematic diagram of the alcoholysis and amino-induced reactions between urushiol (**a**) and organosilane (**b**).

**Table 1 polymers-17-01596-t001:** Substrate specificity and pH optima of the constitutive and inducible forms of laccases isolated from fungal cultures [28].

Somece of Laecase	Form of Laccase	2.6-Dimethoxy Phenol		Ferwlic Acid		Sinapic Acid		Syringic Acid		Vanillic Acid	
		Activity	pH	Activity	pH	Activity	pH	Activity	pH	Activity	pH
*B. cinerea* A235	Constitutive	1000	4.2	570	4.0	1097	3.8	694	4.0	92	4.1
*F. annosus*	Constitutive	1000	4.2	650	4.6	1112	3.0	812	4.3	170	4.9
*F. annosus*	Inducible	1000	4.2	1570	4.6	3430	3.0	2115	4.3	501	4.9
*P. mutabilis*	Constitutive	1000	3.4	420	4.1	1211	3.9	970	4.1	104	4.3
*P. mutabilis*	Inducible	1000	3.4	980	4.1	2870	3.9	2164	4.1	460	4.3
*P. ostreatus*	Constitutive	1000	4.2	514	4.2	1127	3.8	890	4.0	120	4.8
*P. ostreatus*	Inducible	1000	4.2	1374	4.7	3210	3.8	2970	4.0	630	4.8
*P. anserina(−)*	Constitutive	1000	4.4	1460	4.2	2115	4.0	1860	4.2	616	4.5
*R. praticola*	Constitutive	1000	6.8	910	7.2	1970	6.5	1215	7.0	170	7.5
*T. versicolor*	Constitutive	1000	3.8	190	4.0	987	3.6	420	4.0	97	5.0
*T. versicolor*	Inducible	1000	3.8	1028	4.0	2126	3.6	1270	4.0	420	5.0

Note: activity: relative units (normalized to 2,6-dimethoxyphenol = 1000); pH: unitless.

**Table 2 polymers-17-01596-t002:** Physical properties of urushi film [34].

Testing Item	Fi1m of Urushiol Polymer 502, Catalyzed by Cu-D370	Film of Super Urushiol Polymer	Film of Ordinary Urushi
Surface dry (min.)	<20	<30	30–120
Hard dry (day)	1	3	30–90
Luster value (%)	120–130	120–140	57
Appearance	Purple	Purple	Dark brown
Impact strength (Kg × cm)	Front > 50; Back > 50	Front > 50; Back > 50	<30
Adhesion (grade)	1	1–2	4
Hardness (glass value)	0.78	0.82	0.78–0.89

## Data Availability

Data are contained within the article.

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
