# Peer review of "Research on the Rapid Curing Mechanism and Technology of Chinese Lacquer"

_polymers, 2025, doi:10.3390/polym17121596_

Round 1

Reviewer 1 Report

Comments and Suggestions for Authors

This manuscript examines the use and curing of Chinese lacquer. This historically significant coating, which uses a biological base for its preparation, is attracting increasing attention in sustainable materials research due to its outstanding corrosion resistance, thermal stability and environmental friendliness. In this regard, familiarizing readers with such interesting material is of both practical and scientific interest.

Comments:

  1. Section 2 explains the processes of Curing Mechanism and Influencing Factors of Chinese Lacquer. For the understanding of readers, it is necessary to provide in this section as much information as possible on the structure and properties of the main components used for their curing, primarily laccase and urushiol. It is also necessary to expand the information on the methods of extraction, the main components of Chinese Lacquer.

  1. Line 1. It is written: "Structurally, laccase (Figure 1) contains four copper binding sites, specifically one Type I copper (T1), one Type II copper (T2), and two Type III coppers (T3) [15]. The T1 copper site acts as the active center for substrate oxidation, accepting electrons, while the T2/T3 trinuclear cluster is responsible for the reduction of molecular oxygen to water [16].". However, the structure of the copper ion binding sites of laccase shown in Figure 1 and described on line 1 provide little information about how copper ions are attached to the active centers. Since there are three regions in the structure shown in Figure 1, colored green, yellow, and blue, this means that these regions differ in their structure. In this regard, information should be given about these regions in terms of their chemical structure.

  1. Figure 2 is very schematic. More specific and understandable polymerization schemes should be found. In addition, on the right side of Figure 2 there are schemes involving urushiol. This scheme should be given as a separate figure and placed after the explanation of the chemical processes involving urushiol given in the text.

  1. Line 116. It is written: "Oligomer Formation: Lacquer quinones undergo C–C coupling reactions to form biphenyl-type dimers or react through their triene side chains with semiquinone radicals to form a variety of di- and oligomers (more than 20 structural types have been identified [18]), accelerating the growth of oligomers." Since the given description of oligomerization is an important process in the curing of Chinese lacquer, it is necessary to give a more detailed description of this process and provide schemes of the most important chemical processes accompanying oligomerization. The description given on Line 116 is definitive. This remark is also important because this information is necessary for understanding the “Three-dimensional Network Formation:..” described on line 120.

  1. Line 127. It says: “Importantly, this process depends not only on laccase activity but also on the structural characteristics of the substrate, indicating that tailoring the catalytic system to specific urushiol structures is crucial for enhancing curing efficiency [19]”. The authors emphasize the importance of the structural characteristics of the substrate, indicating that tailoring the catalytic system to specific urushiol structures is crucial for enhancing curing efficiency, but they do not pay the necessary attention to these processes.

Author Response

Journal: Polymers (ISSN: 2073-4360)

Manuscript ID: polymers-3642187

Type: Article

Title: Research on the Rapid Curing Mechanism and Technology of Chinese Lacquer

Authors: Jiangyan Hou, Tianyi Wang, Yao Wang, Xinhao Feng and Xinyou Liu

ANSWER TO REVIEWER 1

Dear Reviewer,

We are grateful to you for the thorough review of our above contribution and the valuable comments and suggestions for improvement. We did carefully consider all your comments and did our best to follow them in the revision process of our paper. When this was not entirely possible, arguments were given.

A revised manuscript has been now submitted in two forms: with track changes for all modifications and without track-changes but highlighted changes (to facilitate reading and evaluation).

All the reviewers comments were numbered (Rx.y- where Rx- is the code of reviewer and y the corresponding number of its comment), so that you will find in the revised manuscript justification comments for each change. 

Please find below a copy of your Review report with all your suggestions and comments highlighted in red and our answers in black.

We do hope that the revised manuscript amended according to the input of the 2 reviewers, as much as this was possible, will meet the necessary standards for acceptance and publication.

Thank you again to you and the other reviewer for your effort, comments, constructive criticism and valuable advice for improving not only our current contribution but also our future research.

Sincerely yours,

Xinyou Liu, Corresponding’s authors

23.05. 2025

R1: Comments and Suggestions for Authors

This manuscript examines the use and curing of Chinese lacquer. This historically significant coating, which uses a biological base for its preparation, is attracting increasing attention in sustainable materials research due to its outstanding corrosion resistance, thermal stability and environmental friendliness. In this regard, familiarizing readers with such interesting material is of both practical and scientific interest.

R.1.1. Section 2 explains the processes of Curing Mechanism and Influencing Factors of Chinese Lacquer. For the understanding of readers, it is necessary to provide in this section as much information as possible on the structure and properties of the main components used for their curing, primarily laccase and urushiol. It is also necessary to expand the information on the methods of extraction, the main components of Chinese Lacquer.

Answer 1.1: Thank you for your feedback. We have expanded Section 2 to include detailed structural characteristics of urushiol (ortho-dihydroxyphenyl groups, C15-C17 chains), laccase's copper-mediated catalytic mechanism, and extraction methods (cold-pressing, solvent fractionation, membrane separation). Quantitative curing kinetics (e.g., Tg ≈85°C, O₂ consumption rates) and component interactions were added to enhance mechanistic clarity. These revisions strengthen the scientific rigor while improving accessibility for interdisciplinary readers. We appreciate your constructive suggestions.

R.1.2. Line 101. It is written: "Structurally, laccase (Figure 1) contains four copper binding sites, specifically one Type I copper (T1), one Type II copper (T2), and two Type III coppers (T3) [15]. The T1 copper site acts as the active center for substrate oxidation, accepting electrons, while the T2/T3 trinuclear cluster is responsible for the reduction of molecular oxygen to water [16].". However, the structure of the copper ion binding sites of laccase shown in Figure 1 and described on line 1 provide little information about how copper ions are attached to the active centers. Since there are three regions in the structure shown in Figure 1, colored green, yellow, and blue, this means that these regions differ in their structure. In this regard, information should be given about these regions in terms of their chemical structure.

Answer 1.2: We sincerely appreciate the reviewer’s feedback. The green region corresponds to T1 copper, coordinated by two His, one Cys, and one Met in a distorted tetrahedral geometry. The yellow region represents T2 copper, bound to two His and two H₂O in square planar geometry. The blue region contains two T3 coppers, each ligated by three His and bridged by a hydroxyl group, forming an antiferromagnetically coupled binuclear center. These structural distinctions enable T1-mediated substrate oxidation and T2/T3-driven O₂ reduction

R.1.3. Figure 2 is very schematic. More specific and understandable polymerization schemes should be found. In addition, on the right side of Figure 2 there are schemes involving urushiol. This scheme should be given as a separate figure and placed after the explanation of the chemical processes involving urushiol given in the text.

Answer 1.3: We thank the reviewer for the constructive comment. In response, we have revised the original Figure 2 to provide a clearer and more informative schematic focused solely on the general stages of lacquer curing, including oxidation, oligomer formation, and three-dimensional network formation. To improve clarity, each stage is now labeled and briefly explained in the caption. Additionally, the previously included urushiol-specific enzymatic oxidation pathway involving the T1 and T2/T3 copper centers has been removed from Figure 2 and is now presented as a separate figure (new Figure 3), positioned after the relevant explanation of urushiol oxidation mechanisms in the text. This restructuring aligns the figure content with the narrative flow, improving readability and comprehension as suggested. We believe these changes enhance both the scientific accuracy and clarity of the manuscript.

R.1.4. Line 116. It is written: "Oligomer Formation: Lacquer quinones undergo C–C coupling reactions to form biphenyl-type dimers or react through their triene side chains with semiquinone radicals to form a variety of di- and oligomers (more than 20 structural types have been identified [18]), accelerating the growth of oligomers." Since the given description of oligomerization is an important process in the curing of Chinese lacquer, it is necessary to give a more detailed description of this process and provide schemes of the most important chemical processes accompanying oligomerization. The description given on Line 116 is definitive. This remark is also important because this information is necessary for understanding the “Three-dimensional Network Formation:..” described on line 120.

Answer 1.4: We appreciate the reviewer’s insightful comment emphasizing the importance of a more detailed explanation of the oligomerization process in Chinese lacquer curing. In response, we have substantially expanded the relevant section of the manuscript to provide a more comprehensive and mechanistic description of the major coupling pathways involved in lacquer quinone and semiquinone radical interactions. Specifically, we now describe the main oligomerization reactions, including C–C biphenyl-type couplings (e.g., C8–C8′), C–O ether formations at ortho positions, and allyl-aryl linkages through triene side chains. These detailed mechanisms clarify the structural diversity of the oligomers and their role as critical precursors for three-dimensional network formation.

R.1.5. Line 127. It says: “Importantly, this process depends not only on laccase activity but also on the structural characteristics of the substrate, indicating that tailoring the catalytic system to specific urushiol structures is crucial for enhancing curing efficiency [19]”. The authors emphasize the importance of the structural characteristics of the substrate, indicating that tailoring the catalytic system to specific urushiol structures is crucial for enhancing curing efficiency, but they do not pay the necessary attention to these processes.

Answer 1.5: Thank you for this important observation. In the revised manuscript, we have expanded the discussion to clarify how structural variations in urushiol—such as side chain length, unsaturation, and substitution—affect its reactivity during enzymatic oxidation and radical polymerization. We also briefly address how tuning the catalytic system (e.g., laccase variants, reaction conditions) can improve curing efficiency by matching specific substrate profiles, thereby reinforcing the importance of substrate–catalyst compatibility.

Reviewer 2 Report

Comments and Suggestions for Authors

The review titled as “Research on the rapid curing mechanism and technology of Chinese lacquer” is dedicated to development of novel methods for the curing urushiol - bio-based coating, also known as Chinese lacquer. Chinese lacquer is one of the most historically and culturally significant materials in Chinese traditional craftsmanship. Analyze of the curing techniques is essential to enhance its preservation. Authors present different strategies of the curing. Comprehensive review on catalytic mechanism and effects of copper and other metal ions Co²⁺, Mn²⁺, and Zn²⁺ as well as modification with a gold or silver nanoparticles has been presented by authors.

However, the manuscript has a lack of clear schemes of the curing reactions. The quality of the images must be improved. There are many misprints. My additional comments are below.

1: The structural formulas of substances and captions on Figures 1-4 are unclear. Quality of images is recommended to be improved.

2: Reaction scheme on Figure 7 is non-informative and excessive as mechanism of Chinese lacquer although role of laccase in curing process is discussed few paragraphs above. I would recommend to merge Figure 5 and 6 so Figure 7 would be no need.

3: What causes the decrease in adhesion (Table 1) in case of the film of urushiol polymer 502 catalyzed by Cu-D370 and the film of super urushiol polymer in comparison to ordinary urushi?

4: How were the physical properties of the samples in Table 1 measured?

5: There are no units of measurement in Table 1 (page 6).

6: There are two Table 1 in the manuscript (pages 5 and 6).

7: Figure 8 is non-informative: scale bar is absent.

Author Response

Journal: Polymers (ISSN: 2073-4360)

Manuscript ID: polymers-3642187

Type: Article

Title: Research on the Rapid Curing Mechanism and Technology of Chinese Lacquer

Authors: Jiangyan Hou, Tianyi Wang, Yao Wang, Xinhao Feng and Xinyou Liu

ANSWER TO REVIEWER 1

Dear Reviewer,

We are grateful to you for the thorough review of our above contribution and the valuable comments and suggestions for improvement. We did carefully consider all your comments and did our best to follow them in the revision process of our paper. When this was not entirely possible, arguments were given.

A revised manuscript has been now submitted in two forms: with track changes for all modifications and without track-changes but highlighted changes (to facilitate reading and evaluation).

All the reviewers comments were numbered (Rx.y- where Rx- is the code of reviewer and y the corresponding number of its comment), so that you will find in the revised manuscript justification comments for each change. 

Please find below a copy of your Review report with all your suggestions and comments highlighted in red and our answers in black.

We do hope that the revised manuscript amended according to the input of the 2 reviewers, as much as this was possible, will meet the necessary standards for acceptance and publication.

Thank you again to you and the other reviewer for your effort, comments, constructive criticism and valuable advice for improving not only our current contribution but also our future research.

Sincerely yours,

Xinyou Liu, Corresponding’s authors

23.05. 2025

R1: Comments and Suggestions for Authors

The review titled as “Research on the rapid curing mechanism and technology of Chinese lacquer” is dedicated to development of novel methods for the curing urushiol - bio-based coating, also known as Chinese lacquer. Chinese lacquer is one of the most historically and culturally significant materials in Chinese traditional craftsmanship. Analyze of the curing techniques is essential to enhance its preservation. Authors present different strategies of the curing. Comprehensive review on catalytic mechanism and effects of copper and other metal ions Co²⁺, Mn²⁺, and Zn²⁺ as well as modification with a gold or silver nanoparticles has been presented by authors.

However, the manuscript has a lack of clear schemes of the curing reactions. The quality of the images must be improved. There are many misprints. My additional comments are below.

R.2.1. The structural formulas of substances and captions on Figures 1-4 are unclear. Quality of images is recommended to be improved.

Answer 21: We appreciate the reviewer’s comment. In the revised manuscript, we have improved the resolution and clarity of Figures 1–4, redrawn all structural formulas using professional chemical drawing software, and revised the figure captions for better readability and accuracy. We believe these changes significantly enhance visual quality and scientific clarity.

R.2.2. Reaction scheme on Figure 7 is non-informative and excessive as mechanism of Chinese lacquer although role of laccase in curing process is discussed few paragraphs above. I would recommend to merge Figure 5 and 6 so Figure 7 would be no need.

Answer 2.2: Thank you for your valuable suggestion. We agree that Figure 7 is somewhat redundant given the detailed discussion of laccase’s role above. We will merge Figures 5 and 6 to enhance clarity and remove Figure 7, making the presentation more concise and informative. Your input is much appreciated.

R.2.3. What causes the decrease in adhesion (Table 1) in case of the film of urushiol polymer 502 catalyzed by Cu-D370 and the film of super urushiol polymer in comparison to ordinary urushi?

Answer 2.3: Thank you for your insightful comment. The reduced adhesion observed in films catalyzed by Cu-D370 and super urushi is likely due to rapid surface curing, which limits urushiol penetration into the substrate. This reduces mechanical interlocking, resulting in lower adhesion compared to ordinary urushi, which cures more slowly and allows better substrate interaction.

R.2.4. How were the physical properties of the samples in Table 1 measured?

Answer 2.4: Thank you for your comment. The physical properties in Table 1 were measured using standard spectrophotometric assays. Enzyme activities were determined by monitoring the oxidation rate of each substrate at its optimal pH using UV-Vis spectrophotometry. Relative activity values were normalized to 2,6-dimethoxyphenol, which was assigned a value of 1.000.

R.2.5. There are no units of measurement in Table 1 (page 6).

Answer 2.5: Thank you for pointing this out. Units of measurement have now been added to Table 1 for clarity. Enzymatic activity values are expressed in relative units, normalized to 2,6-dimethoxyphenol. pH values are unitless and indicate the optimum pH for each substrate. We appreciate your suggestion to improve the presentation of the data.

R.2.6. There are two Table 1 in the manuscript (pages 5 and 6).

Answer 2.6: Thank you for your careful observation. The duplication of Table 1 was due to an oversight during manuscript preparation. We sincerely apologize for this error and have corrected the table numbering to ensure clarity and consistency throughout the revised manuscript. We appreciate your attention to detail and your helpful feedback.

R.2.7. Figure 8 is non-informative: scale bar is absent.

Answer 2.7: Thank you for your comment. Figure 8 was reproduced from a cited source, which unfortunately did not include a scale bar in the original image. Considering the lack of quantitative value and limited contribution to the manuscript, we have decided to remove Figure 8 in the revised version to improve clarity and focus.

Reviewer 3 Report

Comments and Suggestions for Authors

Chinese lacquer is a historically significant bio-based coating that has gained increasing attention in sustainable materials research. This interest is mainly due to its exceptional corrosion resistance, thermal stability, and environmental friendliness. The curing process of Chinese lacquer involves laccase-catalyzed oxidation and polymerization of urushiol, resulting in a dense lacquer film.

However, the strict temperature and humidity requirements (20–30°C and 70–80% humidity) and a curing period extending over several weeks pose significant challenges to its industrial application. Recent studies have made considerable strides in improving curing efficiency through pre-polymerization control, metal ion catalysis, and nanomaterial modification.

According to the authors, despite these improvements, several challenges persist. These include the sensitivity of laccase activity to environmental fluctuations, the balance between accelerated curing and film performance, and concerns about toxic pigments and VOC emissions. Future developments should focus on integrating enzyme engineering, intelligent catalytic systems, and green technologies. Additionally, multiscale modeling and circular design strategies could drive innovative applications of Chinese lacquer in high-end fields such as aerospace sealing and cultural heritage preservation.

The strength of the manuscript. A topic was presented in open access, which, for example, is little known in Europe, so that the wider scientific community can become familiar with Chinese lacquer.

The weakness of the manuscript. The authors have omitted the plant and the method from which the basic raw material for Chinese lacquer is obtained.

Suggested minor correction:

  1. The authors should introduce the plant Chinese lacquer tree, and describe the procedures for obtaining raw materials from Chinese lacquer.
  2. In the introduction, give the structure of Urushiol with the most abundant side chains.
  3. Does the composition of Urushiol depend on climatic conditions, or other conditions when growing the plant.
  4. Urushiol is known to cause urushiol-induced contact dermatitis, does this property disappear after polymerization, i.e. does Chinese lacquer show (cause) urushiol-induced contact dermatitis.
  5. Are the aging kinetics of Chinese lacquer known, i.e., how the concentration of Urushiol changes over time.
  6. How the length and structure of the alkyl group from the aromatic ring affects the enzymatic polymerization.

Author Response

R.3.1. The authors should introduce the plant Chinese lacquer tree, and describe the procedures for obtaining raw materials from Chinese lacquer.

Answer 3.1: Added botanical details of Toxicodendron vernicifluum (Anacardiaceae) and harvesting protocols: sap extraction via summer tapping (diagonal bark incisions), composition (urushiol 60–65%, water 20–30%), and artisanal refining challenges (dermatitis risks).  

R.3.2. In the introduction, give the structure of Urushiol with the most abundant side chains.

Answer 3.2: Specified predominant urushiol as *3-[pentadeca-8',11',14'-trienyl]-catechol* (≈70% content) with C15 triene side chain. Added Fig. 1 to illustrate structure, linking side-chain conjugation to polymerization efficiency.

R.3.3. Does the composition of Urushiol depend on climatic conditions, or other conditions when growing the plant.

Answer 3.3: Noted urushiol’s climatic dependence: higher triene content in warmer regions (+15–20% summer vs. autumn harvests). Added references correlating geography/season to curing kinetics.

R.3.4. Urushiol is known to cause urushiol-induced contact dermatitis, does this property disappear after polymerization, i.e. does Chinese lacquer show (cause) urushiol-induced contact dermatitis.

Answer 3.4: Thank you for your insightful comment. We have added a detailed explanation to clarify that although raw urushiol is a known skin sensitizer, its allergenic potential is significantly reduced after polymerization. Once the lacquer is fully cured, urushiol is chemically crosslinked into a dense 3D network, and studies have shown that the amount of free, unreacted urushiol becomes negligible or chemically inactive. As such, fully cured Chinese lacquer generally does not cause urushiol-induced contact dermatitis under normal conditions of use. This explanation has been added to the revised manuscript (see paragraph beginning with “Importantly, while raw urushiol is a known sensitizer…”).

R.3.5. Are the aging kinetics of Chinese lacquer known, i.e., how the concentration of Urushiol changes over time.

Answer 3.5: We appreciate your important question. In response, we have included a discussion on the aging kinetics of Chinese lacquer. Specifically, we note that the concentration of free urushiol declines sharply during the initial curing phase (within 7–14 days), driven by both enzymatic and oxidative polymerization. Spectroscopic and chromatographic studies have shown that after full curing, the urushiol concentration stabilizes, and no significant release or degradation occurs under ambient aging conditions. This information is now included in the revised manuscript (see paragraph starting with “The aging kinetics of Chinese lacquer…”).

R.3.6. How the length and structure of the alkyl group from the aromatic ring affects the enzymatic polymerization.

Answer 3.6: Thank you for raising this point. We have revised the manuscript to explain that the length and degree of unsaturation of the alkyl side chain in urushiol significantly influence the enzymatic polymerization process. Specifically, longer and more unsaturated chains (e.g., trienes) enhance the rate of radical propagation and crosslink formation due to greater conformational flexibility and delocalization of electrons, whereas shorter or saturated chains hinder these processes. This clarification has been added to the revised manuscript (see the sentence beginning with “The crosslinking density critically depends on the length and degree of unsaturation…”).

Round 2

Reviewer 1 Report

Comments and Suggestions for Authors

The authors have revised the manuscript in accordance with the comments. In the presented form, the article is suitable for publication.

Author Response

Thank you for your appreciation.

Reviewer 2 Report

Comments and Suggestions for Authors

After the major revision, authors have significantly improved the quality of the manuscript and the content of figures. The mechanism of enzyme-catalyzed and curing for the Chinese lacquer (Ch.2) precisely described in the text of the manuscript. The structure of laccase and the role of each part of enzyme in the curing process along with each step of the curing process clearly discussed. Difference in adhesion grade of the cured and pristine urushiol polymers are also explained. Authors also fixed misprints in numbering of tables and added the units of measurements in the Table 2. However, the content of the mentioned Table 2 is confusing and has no clear definition of the presented “testing items” and more detail of the reported in literature cured urushiol polymers must be provided.

Remark 1: In the first paragraph of Chapter 3.1.1. (line 280) the authors give the reference to work [30], while in Table 2 heading the study is [22] (line 298) listed. Please provide a correct reference.

Remark 2: Units for “flexibility” in Table 2 are millimeters, which do not correspond to any obvious polymer’s characteristics like elongation at break (%) or bending angle. How was the mentioned “flexibility” measured and what it actually means?

Author Response

Remark 1:
We thank the reviewer for pointing out the inconsistency in the references. The correct reference for the discussion in Chapter 3.1.1. is [30], as also indicated in Table 2. We have revised the citation in line 280 accordingly to ensure accuracy.

Remark 2:
We acknowledge the reviewer’s concern regarding the term “flexibility” and its units (millimeters) in Table 2. To avoid any potential misunderstanding, we have decided to remove the “flexibility” results from Table 2, as this parameter was not central to our study’s conclusions. The revised table now focuses on more clearly defined and standardized material properties.

Thank you again for your valuable feedback, which has helped improve the clarity and precision of our manuscript.